# Heat Acclimation under Drought Stress Induces Antioxidant Enzyme Activity in the Alpine Plant *Primula minima*

**DOI:** 10.3390/antiox12051093

**Published:** 2023-05-13

**Authors:** Thomas Roach, Gilbert Neuner, Ilse Kranner, Othmar Buchner

**Affiliations:** Department of Botany, University of Innsbruck, Sternwartestrasse 15, 6020 Innsbruck, Austria

**Keywords:** heat tolerance, drought stress, reactive oxygen species, climate change, acclimation, antioxidant, ascorbate peroxidase, catalase, glutathione

## Abstract

Heat and drought stresses are increasingly relevant topics in the context of climate change, particularly in the Alps, which are warming faster than the global average. Previously, we have shown that alpine plants, including *Primula minima*, can be gradually heat hardened under field conditions in situ to achieve maximum tolerance within a week. Here, we investigated the antioxidant mechanisms of *P. minima* leaves that had been heat hardened (H) without or with (H+D) additional drought stress. Lower free-radical scavenging and ascorbate concentrations were found in H and H+D leaves, while concentrations of glutathione disulphide (GSSG) were higher under both treatments without any change in glutathione (GSH) and little change in glutathione reductase activity. In contrast, ascorbate peroxidase activity in H leaves was increased, and H+D leaves had >two-fold higher catalase, ascorbate peroxidase and glucose-6-phosphate dehydrogenase activities compared with the control. In addition, the glutathione reductase activity was higher in H+D compared with H leaves. Our results highlight that the stress load from heat acclimation to maximum tolerance is associated with a weakened low-molecular-weight antioxidant defence, which may be compensated for by an increased activity of antioxidant enzymes, particularly under drought conditions.

## 1. Introduction

Small-stature alpine plants maintain their temperature well above the cool ambient air simply by having a stable aerodynamic boundary layer that is resistant to heat exchange [1]. This morphological strategy provides thermal benefits in a cold environment [2,3,4,5], but is counterproductive during heatwaves. As such, intentional overheating can become disadvantageous, leading to heat damage, which has been recurrently observed in small-growing alpine plants on hot days [6,7,8,9,10]. Climate change increases the intensity and frequency of heatwaves, which have tripled across Europe since 1958 [11]. The European heatwave in the summer of 2003 [12] was likely the warmest event since 1540 [13]. At a high elevation in the European Alps, temperature increases doubled the mean global surface increase (+0.55 K) [14], and high elevation sites seem to be particularly affected by global warming [15]. Hence, heat stress must be considered significant for alpine species and the progressive upward migration of alpine plant species is already considered to be a response to the warming climate [16].

Heatwaves are regular climatic components that can occur anywhere in the world and usually consist of a series of consecutive days of high temperatures, clear skies, and a dry atmosphere that successively leads to soil drought [17,18,19]. Drought leads to an increase in heat stress at the plant level as reduced or impaired stomatal conductance lowers transpiration cooling. At alpine sites, overheating is radiation-related, and combines with full solar radiation input, which can be exacerbated by elevation [20]. Together, these environmental factors produce a toxic mix that can pose serious problems to photosynthesis, namely strong irradiation under heat with concomitant drought-induced stomatal closure. As a consequence, alpine species are expected to show a high degree of specialisation towards such stress factor combinations and a specific tolerance to the associated stresses.

Reactive oxygen species (ROS), such as hydrogen peroxide (H_2_O_2_) and singlet oxygen (^1^O_2_), are characterized by a higher reactivity than ground state oxygen (triplet oxygen; ^3^O_2_) that enables them to function in terms of signal transduction, but in excess, they lead to cellular damage. Heat stress can lead to the elevated production of ROS via different pathways. Firstly, during acclimation to sub-critical elevated temperatures, extracellular ROS are propagated from cell-to-cell via membrane-bound NADPH oxidases, as part of systematic stress signalling [21,22,23,24], helping to achieve acquisition of heat tolerance [25,26]. Under extreme heat stress, organelle membranes hosting electron transport chains (chloroplast, mitochondria) become overly fluid and associated protein complexes involved in electron transfer denature, leading to an uncoupling of electron flow from acceptors (e.g., NADP^+^), and such high ROS levels cause exhaustion or inactivation of antioxidant defences [27,28,29]. A variety of antioxidants function collectively to prevent redox imbalances and so-called ‘oxidative stress’. These include enzymes, such as catalases and ascorbate peroxidases that directly metabolize ROS (e.g., H_2_O_2_), low-molecular-weight (LMW) antioxidants, such as ascorbate, tocochromanols and glutathione (GSH) that directly react with ROS or products of ROS-associated damage (e.g., carbonyls from lipid peroxides), and enzymes that recycle oxidised LMW antioxidants after they have reacted with ROS, such as glutathione and dehydroascorbate reductases [30,31,32]. Thus, acclimation to increased temperatures include redox modifications, and responses to heat hardening likely involve a change in antioxidant capacity.

Drought leads to turgor loss, osmotic stress and restrictions in CO_2_ influx as stomata close to prevent water loss. Redox regulatory and antioxidant systems have a paramount role in drought tolerance [33], particularly in C3 plants that do not concentrate CO_2_ in the vicinity of RUBISCO [34]. Regarding ROS signalling, the transcriptional response of *Arabidopsis thaliana* to drought overlaps with the response to elevated ^1^O_2_, as produced by photosystem II (PSII), but not with the response to increased H_2_O_2_ production in the chloroplast [35]. This agrees with measurements that have shown low CO_2_ availability lowering H_2_O_2_ production rates in the chloroplast [36], and increasing catalase activity to break down excess H_2_O_2_ in peroxisomes [37]. Due to limited CO_2_ availability under drought, the oxidase activity of RUBISCO increases, leading to glycolate accumulation (photorespiration) that is broken down by glycolate oxidase in the peroxisome, releasing H_2_O_2_ [34]. The oxidase activity of RUBISCO also increases under heat stress, thus further enhancing photorespiration and negatively impacting plant growth [38,39].

*Primula minima* is a perennial alpine plant inhabiting non-calcareous rock in the Eastern Alps, the Carpathians and the Balkans between elevations of 1500 to 3000 m. It often grows as a pioneer over bare soil or amongst rocks at disturbed locations (Figure 1), rendering the plant susceptible to high localised temperatures [40]. 

Previously, using a purpose-built heat tolerance testing system (HTTS) that is able to control the heat exposure of whole plants under nearly natural field conditions, we were able to increase the critical high temperature threshold of PSII in *P. minima* from 43.1 °C to 46.7 °C under well-watered conditions (H), and up to 48.0 °C when heat hardening was combined with drought (H+D). Moreover, leaf tissue (50% survival; LT_50_) tolerance increased from 45.2 °C to 54.9 °C (H) and 51.6 °C (H+D), after 6–8 days of gradual heat hardening, which was accompanied in both cases by increased ^1^O_2_ scavenging capacity [40]. 

While heat-stress responses were often conducted by placing plants directly under elevated temperatures with no time to acclimate beforehand, the study of Buchner et al. [40] allowed plants to acclimate to elevated temperatures under field conditions, which is physiologically relevant to plant responses to heatwaves. In this study, our aim was to investigate leaf antioxidant mechanisms in heat-acclimated *P. minima* plants, which may help to explain how plants achieve maximal heat tolerance and regulate ROS levels. For this, we quantified LMW antioxidant concentrations, total antioxidant activity, and antioxidant enzyme kinetics in leaves that had been heat hardened, with or without drought, compared with non-heat-hardened leaves.

## 2. Materials and Methods

### 2.1. Plant Material and Heat Hardening

Growth and treatment conditions of the plants are fully described by Buchner et al. [40]. Briefly, the plants were excavated just beneath the summit of Mt. Patscherkofel (N 47°12′35″/E 11°27′45″), Tyrol, Austria, potted and transported to the Alpine Garden Laboratory of the University of Innsbruck on Mt. Patscherkofel at an elevation of 1950 m (a.s.l.), during the summer of 2015. For heat-hardening treatments, plants were placed in a small cold frame made out of transparent Plexiglas panels (Biostar 1500, Juwel H. Wüster GmbH, Imst, Austria). The cold frames were fully temperature regulated (CR1000, AM 25 T, Campbell Scientific, Loughborough, UK). Heating was provided by ceramic heaters (12 V/150 W; AT858, Conrad Electronics, Hirschau, Germany) and cooling by ventilating fresh air via axial fans (120 mm × 120 mm × 38 mm; RD12038B12H, Conrad Electronics, Hirschau, Germany) from outside. Leaf temperatures were measured with fine-wire thermocouple sensors (TT-Ti-40, Omega Engineering Inc., Stamford, CT, USA). Mean leaf temperatures were increased from 32 °C to 38 °C over 8 days prior to the experiments. For treatment without additional drought stress (i.e., H), plants were watered every morning, and for additional drought stress (i.e., H+D) only when actual leaf water potentials, determined in 12 mm × 5 mm discs using a PSΨPRO^TM^ water potential system (Wescor Inc., Logan, UT, USA), had approached a pre-dawn critical value of −3.3 MPa (Ψ_act_). In a pre-test, leaves dehydrated to −3.3 MPa fully survived, whereas below −3.3 MPa, drought damage occurred. The drought-treated plants had significantly lower midday Ψ_act_ on the third day and pre-dawn Ψ_act_ on the fourth day of heat hardening [40]. The chambers were regularly calibrated by osmolality standards of 100, 290 and 1000 mmol kg^−1^.

### 2.2. Biochemical Analyses

Leaves from untreated and from heat-hardened (H, H+D) individuals (*n*~25 for each treatment) were removed at midday, frozen in liquid nitrogen before freeze-drying for 3 days. Leaves from each replicate (*n* = 3–4, as indicated in figure legends) were ground to a fine powder in 2 mL reaction tubes (Eppendorf) with two 2 mm glass using a TissueLyser II (Qiagen, Düsseldorf, Germany) at 30 Hz for 6 min at freezing temperatures. Ground material was stored at −80 °C for 6–12 weeks before conducting measurements.

#### 2.2.1. Total Free Radical Scavenging Activity

The total free radical scavenging activity was determined using 2,2-diphenyl-1-picrylhydrazyl (DPPH) according to the method described by Brand-Williams et al. (1995). For each sample, 10 mg of lyophilized leaf powder were extracted in 1 mL 50% (*v*/*v*) methanol, shaken at 30 Hz for 2 min before centrifugation at 26,000× *g* for 10 min at 4 °C. The supernatant from each sample was transferred into new 2 mL reaction tubes (Eppendorf Hamburg, Germany). Incremental volumes of the supernatant up to 6 µL were pipetted into a 96-well microplate (Bio-Rad, Hercules, CA, USA), made up to 20 µL with 50% (*v*/*v*) methanol, and 200 µL DPPH solution was added (120 µM DPPH in 100% (*v*/*v*) methanol). The decrease in absorbance at 520 nm (i.e., radical scavenging) was followed for 30 min (Synergy HTX, Biotek, Winooski, VT, USA) until the linear decrease in absorbance ceased. Free radical scavenging activity for each sample volume was normalised to its protein content and calculated against a standard curve of ascorbate (0–8 nmols), and averaged.

#### 2.2.2. Spectrophotometric Analysis of Ascorbate

Ascorbate and dehydroascorbate were measured using a colorimetric assay as previously described [41]. For each sample, 25 mg of lyophilized leaf powder was extracted in 2 mL of 6% (*w*/*v*) trichloroacetic acid and centrifuged at 26,000× *g* for 10 min at 4 °C. For the measurement of total ascorbate, 10 µL of supernatant was incubated with 90 µL of 66.7 mM Sørensen’s buffer, pH 7.4, containing 5 mM dithiothreitol (DTT) for 15 min at 25 °C and then with 0.5% (*w*/*v*) N-ethylmaleimide (NEM), after which 100 µL of the working reagent was added. This consisted of reagent A [4.6% (*w*/*v*) trichloroacetic acid, 15.3% (*v*/*v*) orthophosphoric acid, 0.6% (*w*/*v*) FeCl_3_] and reagent B [4% (*w*/*v*) 2,2′-dipyridyl in 70% (*v*/*v*) ethanol] combined in a 11:4 (*v*/*v*) ratio. It was incubated for 5 min at 42 °C before measuring the absorbance at 520 nm. For the measurement of reduced ascorbate, the samples were incubated as for total ascorbate, but without DTT or NEM. For each assay, three technical replicates were measured and averaged for each biological replicate. Amounts were calculated using a standard curve in the range of 0–1 µmol ascorbate, and the amount of dehydroascorbate was calculated by subtracting the amount of reduced ascorbate from total ascorbate. 

#### 2.2.3. HPLC Analysis of LMW Thiols and Disulphides

Low-molecular-weight thiols and disulphides were measured with HPLC as previously described [41]. For each sample, 50 mg of lyophilized leaf powder were extracted in 1.5 mL of 0.1 M ice-cold HCl at 30 Hz for 1 min before centrifugation at 26,000× *g* for 45 min at 4 °C. A total of 120 µL of the supernatant was used for the determination of total LMW thiols and disulphides, and 400 µL for disulphides only. Briefly, the pH was adjusted in both cases to 8.0 by adding 1.5× the sample volume of 200 mM bicine buffer. Total thiol and disulphide concentrations (e.g., GSH + GSSG) were determined after reduction of disulphides by 30 µL of 3 mM DTT, after which thiols were fluorescently labelled with monobromobimane (mBBr) by adding 20 µL of 15 mM mBBr dissolved in acetonitrile. For determination of disulphides only, thiol groups were blocked with 30 µL of 50 mM NEM dissolved in 2-propanol. Excess NEM was removed in a volume of toluene equal to the sample by vortexing, brief centrifugation and removal of the upper phase, and repeated four times. The remaining disulphides in the lower phase were reduced with DTT and labelled by mBBr as above. Labelled thiols were separated using a reverse-phase ODS-Hypersil column (250 mm × 4.6 mm, 3 µm particle size, Bischoff Chromatography, Leonberg, Germany) and detected by fluorescence (excitation: 380 nm; emission: 480 nm), using an Agilent 1100 HPLC system. The concentration of each thiol species was calculated by subtracting the disulphide concentration from the respective total thiols/disulphide amount. The glutathione half-cell reduction potential (*E*_GSSG/2GSH_) was calculated from the molar concentrations of thiols and disulphides considering the leaf water content, and taking into account deviations from standard conditions in terms of pH (7.3) and temperature using the Nernst equation, as described in Schafer and Buettner [42] and shown in Equation (1) below, where *R* is the gas constant (8.314 J K^−1^ mol^−1^), *T* the temperature in K, *n* the number of transferred electrons (i.e., 2) and *F* the Faraday constant (9.6485 × 10^4^ C mol^−1^).
(1)$$E_{\mathrm{GSSG}/2\mathrm{GSH}}=E^{0\mathrm{pH}}-\frac{RT}{nF}\ln\frac{\left[\mathrm{GSH}\right]²}{\left[\mathrm{GSSG}\right]}$$

#### 2.2.4. Antioxidant Enzyme Activities

For each sample, 30 mg of lyophilized leaf powder were extracted in 1.5 mL of 50 mM Sørensen’s buffer, pH 7.5, at 30 Hz for 30 s. The suspension was centrifuged at 4 °C for 5 min at 26,000× *g* and 1 mL of the supernatant was diluted with 1 mL of extraction buffer, avoiding the upper lipid layer, and re-centrifuged for 5 min at 26,000× *g*. Enzymes were purified from interfering LMW compounds with PD10 Sephadex^®^ G-25 desalting columns (GE Healthcare, Chalfont St Giles, UK) using 1.6 mL of the extract and the centrifugation method at 4 °C according to the manufacturer’s protocol. The resulting purified extract was measured immediately. 

Catalase, ascorbate peroxidase, glutathione reductase and glucose-6-phosphate dehydrogenase (G6PDH) activities were measured spectrophotometrically (Synergy HTX, Biotek, Winooski, VT, USA) at 30 °C in 96-well UV-transparent plates (Corning, New York, NY, USA), using a 0.5 cm pathway correction for the Beer–Lambert law. All substrates were dissolved in the extraction buffer and the total reaction volume was 200 µL per well. Catalase activity was measured in 20 μL of extract with 10 mM H_2_O_2_ and following H_2_O_2_ breakdown via the absorbance decrease at 240 nm (ε = 43.6 M^−1^ cm^−1^) between 0.5 to 2 min after combining H_2_O_2_ with the extract. Ascorbate peroxidase activity was measured in 30 µL of extract with 0.5 mM ascorbate and 0.5 mM H_2_O_2_, by following ascorbate consumption via the absorbance decrease at 265 nm (ε = 14,000 M^−1^ cm^−1^) between 0.5 to 2 min after combining the extract with reagents. Glutathione reductase activity was measured in 50 µL extract with 1 mM GSSG and 0.2 mM NADPH, by following NADPH oxidation via the absorbance decrease at 340 nm (ε = 6220 M^−1^ cm^−1^) between 1 to 10 min after combining the extract with reagents. Glucose-6-phosphate dehydrogenase activity was measured in 50 µL extract with 3 mM glucose-6-phosphate and 0.2 mM NADP^+^, by following NADPH formation via the increase in absorption at 340 nm (ε = 6220 M^−1^ cm^−1^) between 1 to 10 min after combining the extract with reagents. Catalytic rates were normalised to protein content of the extract, as measured with the Bradford assay (Bio-Rad, Hercules, CA, USA) in 2 µL of extract. For each assay, three technical replicates were averaged for each biological replicate. 

### 2.3. Statistical Analyses

If homogeneity of variance between replicates was confirmed with Levene’s test, one-way-ANOVA followed by Tukey’s post-hoc test was used for determining significance differences between mean values at *p* < 0.05. When the data were not normally distributed, significance was tested via the Kruskal–Wallis H test, with post-hoc tests between each treatment performed using a Wilcoxon test. Analyses were performed in SPSS Statistics v.21 (IBM, New York, NY, USA).

## 3. Results

### 3.1. Heat Hardening Led to a Loss of Free Radical Scavenging Activity, Irrespective of Drought

To assess the free radical scavenging activity of the leaves, the DPPH test was used, which has been described as an indicator of total LMW antioxidant activity [43]. Leaf extracts of H and H+D plants quenched the DPPH radical by ca. 30% less than leaf extracts from control plants (Figure 2), indicating that the heat-hardening process led to a loss of LMW antioxidants able to scavenge radicals. 

### 3.2. Heat Hardening Led to a Loss of Ascorbate, Irrespective of Drought

Leaf concentrations of reduced ascorbate and dehydroascorbate without heat hardening were 0.88 and 0.18 µmol g^−1^ DW, respectively. In H and H+D leaves, concentrations of reduced ascorbate significantly dropped to <0.6 µmol g^−1^ DW, while dehydroascorbate concentrations were insignificantly different (Figure 3).

### 3.3. Drought Had a Modulating Effect on Glutathione Concentrations during Heat Hardening

Without controlled heat hardening, leaves of *P. minima* contained 0.50 µmol g^−1^ DW total glutathione, of which 0.16 µmol g^−1^ DW was present as GSSG and 0.34 µmol g^−1^ DW as GSH (Figure 4A). Heat hardening increased the concentrations of GSSG, particularly in H+D leaves, which also had lower GSH concentrations, indicative of GSH oxidation to GSSG. This led to H+D leaves having more oxidising (less negative) values of *E*_GSSG/2GSH_ relative to control and H leaves (Figure 4A). The induction of GSH synthesis in response to heat hardening was observed by the increased concentrations of γ-glutamyl-cysteine under drought (Figure 4B), conditions under which GSH concentrations were lowest (Figure 4A). As the *E*_GSSG/2GSH_ is calculated via the Nernst equation, a rise in temperature of 28 °C (i.e., typical difference between night and day) positively shifted the *E*_GSSG/2GSH_ by 10.6–11.7 mV towards more oxidising values (Figure 4B).

### 3.4. Heat Hardening Increased Antioxidant Enzyme Activities, Especially under Drought

Non-heat-hardened plants had catalase, ascorbate peroxidase, glutathione reductase and G6PDH activities of 6.0 ± 0.6, 12.6 ± 1.5, 0.40 ± 0.03 and 0.03 ± 0.01 µkat mg^−1^ protein, respectively. Ascorbate peroxidase activity was 2.2-fold higher in H leaves, and 2.4-fold higher H+D leaves, compared to non-heat-hardened plants (Figure 5). Catalase and G6PDH activities were two-fold higher in H+D leaves, but insignificantly different in H leaves, compared with non-heat-hardened plants (Figure 5). Glutathione reductase activity was significantly higher in H+D leaves compared with H leaves, but after H+D, they were indifferent to the activity in non-heat-hardened plants (Figure 5).

## 4. Discussion

### 4.1. Acclimation during Heat Hardening Involves Redox Signalling

As sessile organisms, plants have remarkable abilities to tolerate highly contrasting conditions. This ‘plasticity’ occurs via acclimation to the changed environment and alters many aspects of physiology and metabolism. Specifically regarding heat tolerance, some examples include the production of heat-shock proteins and changes in membrane composition to counteract the differences in membrane fluidity [44]. Thus, for plant tissue to achieve maximum tolerance towards adverse conditions, this requires time for acclimation. In this study, we slowly increased temperatures over days that allowed *P. minima* acclimation time to achieve maximal thermal tolerance [40]. Furthermore, in natural environments, plants almost always encounter combined environmental stresses. For example, due to climate change, summer heatwaves are an increasing phenomenon [18], and accompanying elevated temperatures lead to rapid drying of the soil and onset of drought. In several model plants, combined heat and drought have been shown to induce unique responses that differ from either heat or drought stress alone [23,45,46,47]. Some of our data show that the response in H+D leaves was different to those in H leaves (e.g., Figure 4 and Figure 5), as discussed below. Nonetheless, responses to one stress factor can accelerate response time and acclimation in local and distal (‘systemic’) tissue to other stresses, in a signal transduction process that involves H_2_O_2_ [23]. Previously, we showed that in *P. minima* and another alpine plant, *Senecio incanus*, leaves respond to heat hardening by drastically lowering jasmonic acid (JA) levels [40]. Zandalinas et al. [23] showed that the allene oxide synthase (*aos*) mutant of *Arabidopsis thaliana* deficient in JA signalling had accelerated H_2_O_2_ systemic signal propagation in response to heat stress. We therefore presume that H_2_O_2_ signalling in *P. minima* and *S. incanus* is enhanced during acclimation to heat stress, for which antioxidants also play a direct regulatory role. 

### 4.2. Heat-Hardened Leaves Had Weakened LMW Antioxidant Defences

The association of ROS production and redox signalling during heat [23,26] and drought stresses [33] indicates that redox regulation is particularly critical when both stresses combine. Maintaining redox homeostasis central to basal metabolic processes involves the LMW compounds ascorbate and glutathione [31]. As antioxidants, they further provide tolerance to stress factors that would otherwise lead to critical imbalances in the cellular redox state [48,49]. Reacting with ROS converts antioxidants to their oxidised forms, and in response to heat hardening, decreases were observed in both ascorbate and GSH (Figure 3 and Figure 4), indicative of a high ROS load in leaves. This likely explains the equal relative decrease in free-radical scavenging of the H and H+D leaves (Figure 2). Ascorbate and GSH partially function together since GSH is a substrate for dehydroascorbate reductase to recycle dehydroascorbate back to ascorbate. Thus, a limitation of GSH may influence ascorbate recycling, while ascorbate may also have been consumed by violaxanthin de-epoxidase in the xanthophyll cycle, which was active during heat hardening [40], and for recycling oxidised α-tocopherol [50,51]. Despite some loss of ascorbate, there was no evidence for a build-up of dehydroascorbate during heat hardening, indicating efficient recycling. However, heat-hardened leaves did accumulate GSSG, particularly under drought (Figure 4B), inducing an oxidative shift in thiol-based cellular redox state (i.e., *E*_2GSH/GSSG_) and reflecting the oxidative load in leaves.

Modulations of the redox state of glutathione (i.e., *E*_2GSH/GSSG_) are part of the decision process of cell fate, on the one hand regulating growth and differentiation, while on the other hand inducing cell death when the *E*_2GSH/GSSG_ shifts to severely oxidising conditions [42,48,52,53]. Relevant in the context of heat tolerance is the influence of temperature on *E*_2GSH/GSSG_, which becomes more oxidising at higher temperatures ([42]; Figure 4A). This could lead to cell fate changing due to a shift in the cellular redox environment towards more oxidizing values under heat stress. In comparison to other studies [48,54,55], *E*_2GSH/GSSG_ was more positive (oxidised), even in the control plants. Due to the harsh environmental conditions (i.e., high light intensity and contrasting day/night temperatures) this may be a reflection of the stressful habitat of such plants. An alternative explanation is that leaves of *P. minima* were compartmentalising GSSG in the vacuole [56]. The *E*_2GSH/GSSG_ would then be lower (more reducing) in other cellular compartments in accordance with their lower GSSG concentrations. Nevertheless, the accumulation of GSSG in H+D leaves indicates that *P. minima* was particularly suffering from heat hardening under drought. In the alpine species *Soldanella alpina* and *Ranunculus glacialis* depletion of GSH rendered leaves sensitive to heat stress, especially under high light [57]. Indeed, H+D leaves were synthesising GSH, as evident by the increased concentration of γ-glutamyl-cysteine (Figure 4B). It has been shown that the plant hormone salicylic acid (SA) is positively associated with glutathione levels [58]. Therefore, activation of glutathione synthesis could be linked to the increase in SA of heat-hardened leaves of *P. minima* [40].

### 4.3. Heat-Hardened Leaves had Enhanced Enzyme Antioxidant Defences

In contrast to the weakened LMW antioxidant defences, heat hardening increased antioxidant enzyme activity. The increased ascorbate peroxidase activity in H+D and H leaves (Figure 5), indicated that heat tolerance required elevated control of H_2_O_2_ production. Due to limited CO_2_ availability under drought, the oxidase activity of RUBISCO increases, leading to glycolate accumulation (photorespiration) that is broken down by glycolate oxidase in the peroxisome, releasing H_2_O_2_ that is broken down in the peroxisome by catalase [59], explaining why catalase activity only increased in H+D leaves (Figure 5). Glutathione reductase activity was the least affected by heat hardening, despite the accumulation of GSSG. This could further indicate sequestration of GSSG in the vacuole, since levels of GSH remained stable. The increase in G6PDH activity in H+D leaves further supports an enhanced need for this pathway in supplying NADPH, as required by glutathione reductase.

## 5. Conclusions

This study has highlighted that specific antioxidant enzymes increased in activity in response to heat hardening, indicating they have a prominent role in heat stress tolerance. Increasing the abundance of ascorbate peroxidase may facilitate increasing heat tolerance in crops, while catalase and G6PDH can be considered for additional drought tolerance that often occurs during heatwaves. More work is required to identify the particular isoforms and corresponding genes that, if upregulated by gene editing, could be helpful to produce more resilient crops in a changing climate.

## Figures and Tables

**Figure 1 antioxidants-12-01093-f001:**
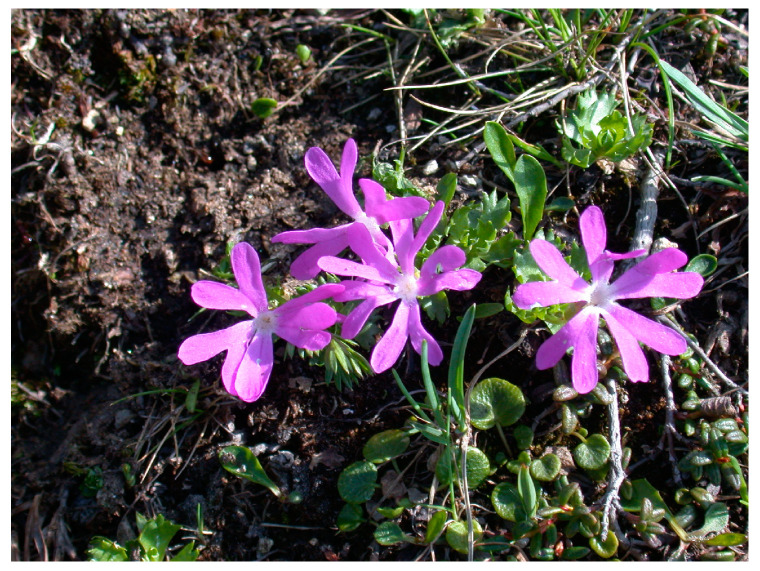
*Primula minima* flowering in the Eastern Alps of Tyrol, Austria.

**Figure 2 antioxidants-12-01093-f002:**
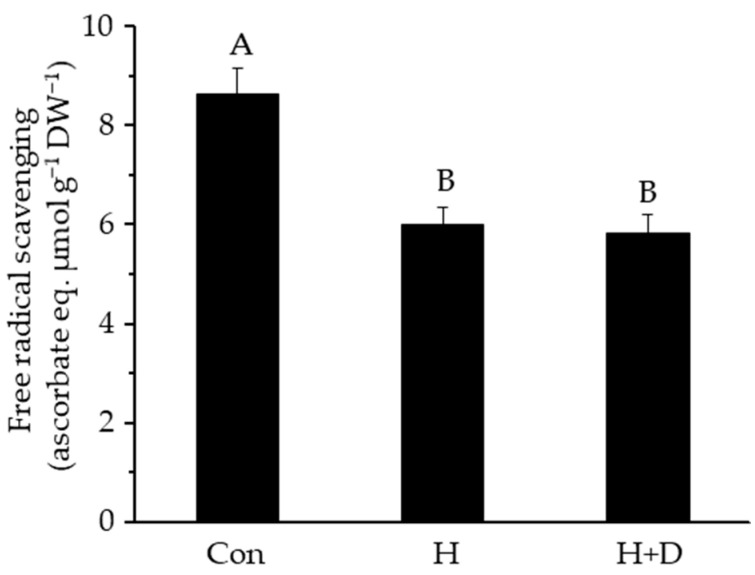
Free radical scavenging activity of leaves. Plants were exposed to increased heat (H) or to increased heat and drought (H+D) and leaves taken for analysis after 8 d of heat hardening, or from non-heat-hardened plants (Con). Free radical scavenging activity was measured using the DPPH assay, normalised to sample dry weight (DW) and expressed as ascorbate equivalents (eq.). Different letters denote significant differences (*p* < 0.05), *n* = 3 ± SD.

**Figure 3 antioxidants-12-01093-f003:**
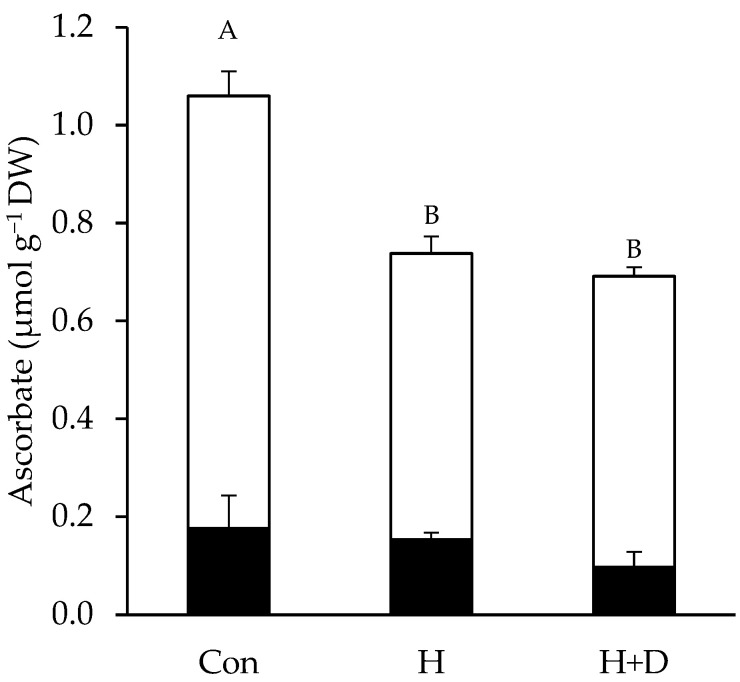
Leaf ascorbate concentrations. Plants were exposed to increased heat (H) or to increased heat and drought (H+D) and leaves taken for analysis after 8 d of heat hardening, or from non-heat-hardened plants (Con). The stacked bar chart shows reduced ascorbate (white bars) above dehydroascorbate (black bars), normalised to sample dry weight (DW). Different letters denote significant differences (*p* < 0.05), *n* = 3 ± SD.

**Figure 4 antioxidants-12-01093-f004:**
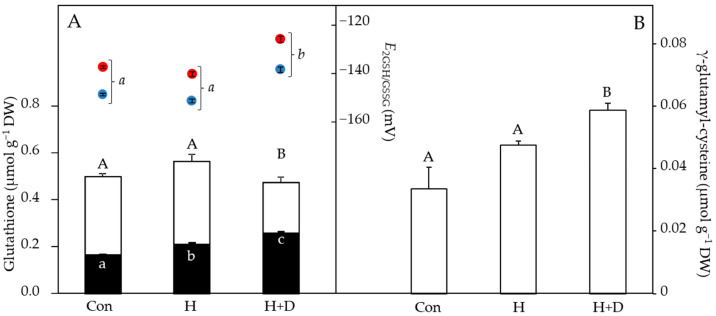
LMW thiol and disulphide concentrations, normalised to sample dry weight (DW), and cellular glutathione redox potential (*E*_GSSG/2GSH_) of leaves in response to heat hardening. Plants were exposed to increased heat (H) or to increased heat and drought (H+D), and leaves taken for analysis after 8 d of heat hardening, or from non-heat-hardened plants (Con). (**A**) The stacked bar chart shows GSH (white bars) and GSSG (black bars), with *E*_GSSG/2GSH_ values above (right y axis), considering daytime (38 °C, red) or night-time (10 °C, blue) temperatures (see methods). (**B**) γ-glutamyl-cysteine concentrations. Significant differences (*p* < 0.05) are denoted by different capital, lower-case and lower-case italic letters for thiols, GSSG and *E*_GSSG/2GSH_, respectively, *n* = 4 ± SD.

**Figure 5 antioxidants-12-01093-f005:**
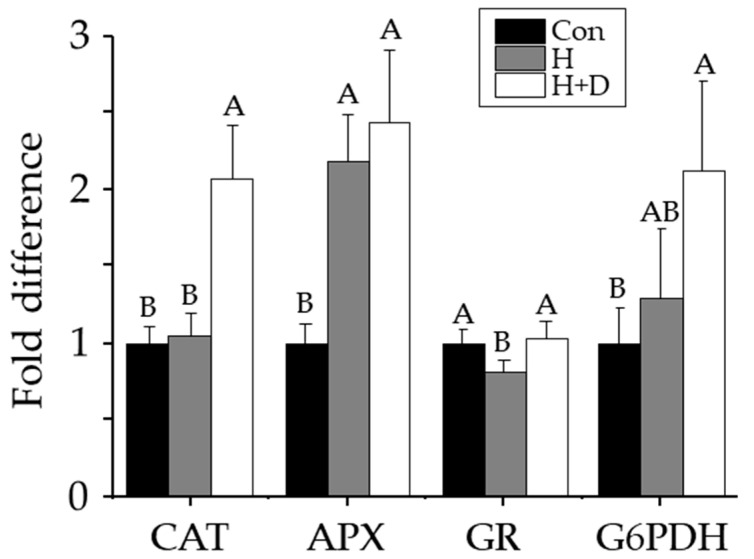
Changes in leaf antioxidant enzyme activities in response to heat hardening. Plants were exposed to increased heat (H; grey) or to increased heat and drought (H+D; white), and leaves taken for analysis after 8 d of heat hardening, or from non-heat-hardened plants (Con; black). Activities of catalase (CAT), ascorbate peroxidase (APX), glutathione reductase (GR) and glucose-6-phosphate dehydrogenase (G6PDH) are shown as fold difference relative to Con levels. Significant differences (*p* < 0.05) calculated individually for each enzyme are denoted by different letters, *n* = 3 ± SD.

## Data Availability

Raw data are available upon request from the corresponding author.

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
