# Peer review of "Heat Acclimation under Drought Stress Induces Antioxidant Enzyme Activity in the Alpine Plant Primula minima"

_antioxidants, 2023, doi:10.3390/antiox12051093_

Round 1
Reviewer 1 Report
Dear Corresponding Author
I checked your paper. It has enough novelty and good information in the related field. But I have some questions that help to improve the paper:
1) In Figures 2-5, about statistical analysis you used ANOVA? Which algorithm you used to compare the to each other? Did you check that your data follow normal distribution? Please clarify these questions to me.
2) There are so many errors in the references list related to format of the journal. Please check and correct them.
Regards
Author Response
Thank you for your review.
For point 1) we have added to the methods: "One-Way-Anova followed by Tukey post hoc test was used for determining significance differences between mean values at P<0.05 after testing for homogeneity of variance using Levene’s test. When data was not normally distributed, significance was tested via the Kruskal–Wallis H test, with post-hoc tests between each treatment performed using a Wilcoxon test. Analyses were performed in SPSS-Statistics v.21 (IBM, New York, NY, USA)."
For point 2) we have revised the references according to MDPI requirements
Reviewer 2 Report
Manuscript (antioxidants-2350069) “A Prominent Role for Enzymes in the Antioxidant Defences of 2 the Alpine Plant Primula minima after Long Term 3 Heat-acclimation under Drought” by Roach et al. presents an interesting study about drought resistance and gene expression analysis in Primula minima.
This manuscript presents a valuable study with breeding applications. However, this manuscript is confused in some parts mainly in the description of the methodology and the experimental design. For these reasons, this manuscript is acceptable for publication in Antioxidants Journal after a major/moderate revision.
The major points for the revision of the manuscript are:
Objectives are very large. Authors must simplify the objectives not including any methodological reference in the separate paragraph.
Phenotyping of drought response must be clarified in the Methodology section. A new figure with the experimental design should be of great interest.
Description of Biochemical analysis is very poor. Authors must detail the protocol.
Figures 2 and 3 should be merged in a single figure
In the Conclusion section, authors must indicate the main implications of these results from an agronomical and breeding point of view.
Author Response
Thank you for your review of our work.
In response to the comments and points raised we have modified the manuscript as follows:
1) we added a separate paragraph with the aims clearly stated, as follows.
"While most studies have investigated heat-stress responses by placing plants directly under elevated temperatures with no time to acclimate beforehand, the study of Buchner et al., [40] allowed plants to acclimate to elevated temperatures under field conditions. Therefore, in this study, our aim was to investigate leaf antioxidant mechanisms in heat-acclimated P. minima plants, which may help explain how plants achieved maximal heat tolerance and regulate ROS levels. For this, we quantified LMW antioxidant concentrations, total antioxidant activity, and antioxidant enzyme kinetics, in leaves that had been heat-hardened with or without drought, and made comparisons to non-heat-hardened leaves"
2) for drought phenotyping in the methods we added the leaf disc dimensions that were measured and number of samples, as well as calibration method. The water potential data during the heat-hardening was already published in Buchner et al., (2017). We now cite this paper and added to the methods when drought induced a significant difference in leaf water potentials to non drought plants during heat hardening.
3) The biochemical methods were fully revised and more precise details provided
4) We disagree that Figures 2 and 3 should be merged because free-radical scavenging assesses more than just ascorbate activity. It would also be inconsistent to add ascorbate data, but not glutathione data since both are LMW antioxidants.
5) We agree this would improve the manuscript and added the following conclusion:
“This study has highlighted that specific antioxidant enzymes increased in activity in response to heat-hardening, indicating they have a prominent role in heat stress tolerance. Increasing the abundance of ascorbate peroxidase may facilitate increasing heat tolerance in crops, while catalase and G6PDH can be considered for additional drought tolerance that often occurs during heat waves. More work is required to identify the particular isoforms and corresponding genes that if gene-edited to be upregulated could provide enhanced yields with regard to our changing climate.”
Reviewer 3 Report
This manuscript describes the function of antioxidant enzymes in heat acclimation of the alpine plant Primula minima. Heat and drought treatments have been performed in the field and evaluated for antioxidant activity. The results show that heat treatment increases the activity of several antioxidant enzymes, whereas radical activity derived from low molecular weight antioxidants is reduced. The original method of heat and drought treatment in the field provides novel data, but the description of the research objectives and discussion in the paper needs extensive revision, and the revised manuscript needs to be re-evaluated.
Major points
1) The title should clearly express the main finding of the paper. It is not clear what "Prominent Role" means. It has been shown that the activity of some antioxidant enzymes increases while the activity of radicals derived from small-molecule antioxidants decreases. However, biochemical and genetic studies are needed to discuss whether antioxidant enzyme activities have a "Prominent Role" or not. For example, the title should be revised to "Heat-acclimation with drought stress induces antioxidant enzyme activity in the alpine plant Primula minima".
2) This paper's purpose is unclear; in the paragraph starting at line 90, it should state what was unexplored in Buchner et al. 2017 and the issues to be addressed in this study.
3) The Discussion contains statements that have little relevance to the results obtained in this paper. For example, 4.1 only mentions the findings of previous studies by the authors and other groups; if ROS signaling is presumed to be induced as described in 4.1, it should be stated in the introduction related to this study's objectives.
4) In line 312, it is stated that "indicative of a high ROS load in leaves." In this discussion, ROS levels, such as hydrogen peroxide in leaves, should be measured.
5) Section 4.4 discusses genome editing for crop breeding. This discussion has little relevance to the results of this paper. Section 4.4 should be omitted because this paper did not find any genes that should be targeted for genome editing in heat acclimation.
Minor editing of English language required. Some description is difficult to read. Please polish the entire paper.
For examples,
Line 240: Does "Without controlled heat-hardening leaves" mean "control leaves without heat-hardening"?
Author Response
Thank you very much for your constructive review of our manuscript. Please find our response to your comments below.
1) we agree the title could be improved and changed it to "Heat-acclimation under drought stress induces antioxidant enzyme activity in the alpine plant Primula minima."
2) We have stated the aims of the study in a new last paragraph of the introduction. "While most studies have investigated heat-stress responses by placing plants directly under elevated temperatures with no time to acclimate beforehand, the study of Buchner et al., [40] allowed plants to acclimate to elevated temperatures under field conditions. Therefore, in this study our aim was to investigate leaf antioxidant mechanisms in heat-acclimated P. minima plants, which may help explain how plants achieved maximal heat tolerance. For this, we quantified LMW antioxidant concentrations, total antioxidant activity, and antioxidant enzyme kinetics, in leaves that had been heat-hardened with or without drought, and made comparisons to non-heat-hardened leaves"
3 & 4) We now mention ROS in the aims (for exact text see response to point 1). We avoided measuring ROS directly because reliably measuring levels in the species investigated is next to impossible. Although many people measure ROS levels, such methods are thwart with difficulties and true intracellular concentrations are likely to be qualitative at best. Moreover, unlike LMW antioxidant concentrations and enzyme kinetics that show more stability over the minutes during the time it takes to sample material from the field, ROS concentrations can fluctuate very rapidly. This has been shown in the studies with fluorescent H2O2 reporter lines in Arabidopsis thaliana, and we think this is relevant to cite to the discussion.
5) This is a relevant point, but we would still like to include this part since heat stress is an increasingly relevant topic for crops too. We would like to leave this for the editor to make the final decision.
Minor: We have reread the paper and made several modifications to help improve readability.
Round 2
Reviewer 1 Report
Dear Corresponding Author
I checked your paper and I think it is now suitable for publication.
Congratulations.
Author Response
Thankyou
Reviewer 2 Report
Authors have revised correctly the manuscript
Author Response
Thank you
Reviewer 3 Report
The revised manuscript addresses and improves upon the issues raised in the previous review. As for Section 4.4, the reviewer has rated it distant from the main content of this paper, but this does not prevent acceptance of the paper. It is up to the editor to decide whether Section 4.4 should be omitted or not.
Author Response
Thank you